# Genome-Wide Transcriptomic and Proteomic Exploration of Molecular Regulations in Quinoa Responses to Ethylene and Salt Stress

**DOI:** 10.3390/plants10112281

**Published:** 2021-10-25

**Authors:** Qian Ma, Chunxue Su, Chun-Hai Dong

**Affiliations:** Shandong Province Key Laboratory of Applied Mycology College of Life Science, Qingdao Agricultural University, Qingdao 266109, China; 15568017741@163.com

**Keywords:** abiotic stress, ethylene, quinoa, proteome, salt stress, transcriptome

## Abstract

Quinoa (*Chenopodium*
*quinoa* Willd.), originated from the Andean region of South America, shows more significant salt tolerance than other crops. To reveal how the plant hormone ethylene is involved in the quinoa responses to salt stress, 4-week-old quinoa seedlings of ‘NL-6′ treated with water, sodium chloride (NaCl), and NaCl with ethylene precursor 1-aminocyclopropane-1-carboxylic acid (ACC) were collected and analyzed by transcriptional sequencing and tandem mass tag-based (TMT) quantitative proteomics. A total of 9672 proteins and 60,602 genes was identified. Among them, the genes encoding glutathione S-transferase (GST), peroxidase (POD), phosphate transporter (PT), glucan endonuclease (GLU), beta-galactosidase (BGAL), cellulose synthase (CES), trichome birefringence-like protein (TBL), glycine-rich cell wall structural protein (GRP), glucosyltransferase (GT), GDSL esterase/lipase (GELP), cytochrome P450 (CYP), and jasmonate-induced protein (JIP) were significantly differentially expressed. Further analysis suggested that the genes may mediate through osmotic adjustment, cell wall organization, reactive oxygen species (ROS) scavenging, and plant hormone signaling to take a part in the regulation of quinoa responses to ethylene and salt stress. Our results provide a strong foundation for exploration of the molecular mechanisms of quinoa responses to ethylene and salt stress.

## 1. Introduction

Quinoa (*Chenopodium quinoa* Willd.), a dicotyledonous plant in the Chenopodiaceae family, originated from the Andean region of South America and has been cultivated for about 7000 years [1]. Quinoa is an allotetraploid plant (2n = 4X = 36), deriving from the genome fusion of two related parent species in the same genus [2]. Quinoa has five major ecotypes depending on its origin centers, including Highlands originating from Peru and Bolivia; Inter-Andean valleys originating from Bolivia, Colombia, Ecuador, and Peru; Salares originating from Bolivia, Chile and Argentina; Yungas originating from Bolivia; and Lowlands originating from Chile [3].

Quinoa has been considered as a pseudo-cereal because of its grain characteristics [4]. Consumption of seeds is the most common use of quinoa. In recent years, quinoa seeds have been reported to have an exceptional balance between oil, protein, and carbohydrate [4,5]. The absence of gluten in its starch allows the development of specific foods for celiac patients [1]. Quinoa seeds are also good sources of vitamins, oil with high linoleate and linolenate content, natural antioxidants, dietary fiber, and minerals [6]. As a result, consumption of quinoa in human diet leads to lower weight gain, improved lipid profile, decreased blood glucose, and increased antioxidant intake [7,8].

In addition to nutritional value in its seeds, quinoa also has resistance to multiple abiotic stresses, including drought, cold and salinity, which allows quinoa to be cultivated in non-native areas [9,10]. As a result, the United Nations Food and Agricultural Organization (FAO) declared quinoa as a major crop for global food security and sustainability under the current climate change conditions and also announced 2013 as the International Year of Quinoa, which is one of only three plants that have received such a designation [9].

Soil salinity is a major abiotic stress, and about 6.5% of the total land area of the world is affected by salt stress [11]. Soil salinization is expected to be amplified, with environmental pollution aggravating, urbanization accelerating, and global warming intensifying it. High salinity in soil causes many damages to plants, including photosynthesis reduction, membrane denaturalization, nutrient imbalance, osmotic imbalance, ion toxicity, stomatal closure, metabolism disruption, reactive oxygen species (ROS) accumulation, and oxidative stress aggravation [12,13].

Plants have evolved various mechanisms to survive high soil salinity. For instance, plants improve their salt tolerance by efficiently controlling Na^+^ sequestration in vacuoles, low cytosolic Na^+^ maintenance, better K^+^ retention, and a high rate of H^+^ pumping to keep an optimal K^+^/Na^+^ ratio in leaves or roots [14]. Organic solutes including proline, soluble sugar, glycine betaine, and polyamines are accumulated to maintain cell turgor under stress [15,16]. In addition, salt stress is able to activate antioxidant enzymes including superoxide dismutase (SOD), peroxidase (POD), and catalase (CAT) to scavenge ROS in responses to soil salinity [17,18].

Plant hormones including abscisic acid (ABA), ethylene, jasmonic acid (JA), and gibberellic acid (GA) play important roles in plant responses to salt stress. ABA was reported to play key roles in plant salt responses [19]. The negative regulator PP2Cs in ABA signaling were proved to confer resistance to salt stress [20]. Several PP2Cs were found significantly up-regulated under salt stress in quinoa [21]. Ethylene also functions in plant responses to salt stress. For example, the *Arabidopsis* ethylene insensitive mutants *ethylene response1-1* (*etr1-1*), *etr2-1*, *ethylene insensitive2-1* (*ein2-1*), *ein2-5*, *ein3-1*, and *ein4-1* showed reduced salt tolerance [22,23,24], while the tobacco and wheat mutants with reduced ethylene sensitivity exhibited increased salt resistance [25,26]. It was reported that ethylene was important for salt responses of *Arabidopsis*, grapevines, maize, and tomato, and ethylene-regulated salt responses in plants mainly by maintaining the homeostasis of Na^+^/K^+^ ratio, nutrients, and ROS, and the assimilation of nitrates and sulfates [27].

Quinoa can survive in moderate to high concentrations of salt, ranging from 150 mM to 750 mM sodium chloride (NaCl), so it is more salt tolerant than wheat, corn, barley, rice, and peas, which show decreased yields when the soil solution exceeds 40 mM NaCl [28]. However, the optimal NaCl concentration for quinoa is between 100 mM to 200 mM [28,29]. Although quinoa shows more salt tolerance than other crops, the molecular mechanism of salt responses in quinoa is largely unknown.

High-throughput genomic transcriptomic analysis provides a way to excavate molecular mechanisms of salt responses in quinoa in the genomic scale [10,21,30,31]. The genome sequence of a quinoa inbred line with an estimated 967 Mb size was fully sequenced in 2017, which provided a reference genome for later transcriptomic analysis [32]. In 2007, the salt tolerance related transmembrane protein coding genes were identified in quinoa by integrating physiological data, RNA-seq, and single nucleotide polymorphism (SNP) analyses [31]. The molecular mechanisms of salt tolerance in epidermal bladder cells of quinoa were also uncovered by RNA-seq [10]. Salt-induced genes were also identified in quinoa treated with 300 mM NaCl for 1 h and 5 d [30]. The differentially expressed genes (DEGs) compared between a salt-tolerant genotype and salt-sensitive genotype were analyzed after treatment with 300 mM NaCl for 0, 0.5, 2, and 24 h in the roots of quinoa seedlings [21]. Considering the importance of the stress-regulated proteins, detection of the differentially expressed changes in protein levels of quinoa is more valuable and practical. However, there have still been no reports about the molecular regulation of salt responses detected by proteomics in quinoa.

To study the molecular mechanism of ethylene-regulated salt responses in quinoa, 4-week-old quinoa seedlings of ‘NL-6′ treated with water, NaCl, and NaCl with 1-aminocyclopropane-1-carboxylic acid (ACC) were collected and analyzed by transcriptional sequencing and tandem mass tag-based (TMT) quantitative proteomics in this research. The DEGs and differentially expressed proteins (DEPs) were analyzed by Gene Ontology (GO) enrichment analysis, Kyoto Encyclopedia of Genes and Genomes (KEGG) enrichment analysis, and their correlation analysis. The molecular regulations of quinoa responses to ethylene and salt stress were analyzed in this research.

## 2. Materials and Methods

### 2.1. Plant Material Treatment and Sample Collection

Seeds of a highland ecotype quinoa, ‘NL-6′, were obtained from Dr. Feng Li of BellaGen (Jinan, China). In this research, 4-week-old quinoa seedlings of ‘NL-6′ treated with water, 300 mM NaCl, and 300 mM NaCl with 100 μM ACC for 0, 3, 6, 9, 12, 24, and 36 h, respectively [21,30,33]. The solutions were directly irrigated to the seedling roots until soil was fully saturated, and then the excess solutions were poured out. The treated plants were collected and frozen in liquid nitrogen. The samples treated for 24 h were used for transcriptional sequencing and TMT quantitative proteomics [33]. The abbreviations of materials used in the transcriptome and proteome are presented in Table 1. The whole seedlings treated for 0, 3, 6, 9, 12, 24, and 36 h were used for later qRT-PCR verification.

### 2.2. Transcriptome Sequencing and Data Analysis

In this research, three independent biological replicates were used, and at least three whole quinoa seedlings were mixed in each replicate. Total RNA was extracted and purified using poly-T oligo-attached magnetic beads. cDNA was synthesized, and adaptors with hairpin loop structures were ligated to prepare for hybridization. The samples were then clustered on a cBot cluster generation system using TruSeq PE Cluster Kit v3-cBot-HS (Illumia). After cluster generation, the library preparations were sequenced on an Illumina Novaseq platform by Novogene Bioinformatics Technology Co. Ltd. (Beijing), and 150 bp paired-end reads were generated. The raw data of FASTQ format were uploaded to the NCBI Sequence Read Archive (SRA), and the SRA accession number is PRJNA726352.

The reference genome was downloaded from the website https://www.ncbi.nlm.nih.gov/genome/?term=quinoa (accessed on 30 June 2022), and the paired-end clean reads were aligned to the reference genome using Hisat2 v2.0.5. Fragments per kilobase of transcript sequence per million (FPKM) of each gene were calculated to estimate gene expression levels based on the length of the gene and reads count mapped to the gene. The genes with corrected *p*-value < 0.05 and absolute fold change ≥2 were considered as significant DEGs. The DEGs were then analyzed by GO enrichment analysis and KEGG enrichment analysis to predict their functions.

### 2.3. Protein Extraction, TMT Labeling, and Proteomics Analysis

In this research, three independent biological replicates were used for protein analysis, and at least three whole quinoa seedlings were mixed in each replicate. The proteomics analyses were performed by Novogene Bioinformatics Technology Co. Ltd. (Beijing, China). In detail, total proteins were extracted by the cold acetone method, labeled by TMT tags, and fractionated using a C18 column on a Rigol L3000 HPLC system. Shotgun proteomics analyses were then performed using an EASY-nLC^TM^ 1200 UHPLC system (Thermo Fisher, Waltham, MA, USA) coupled with a Q Exactive^TM^ HF-X mass spectrometer (Thermo Fisher) operating in the data-dependent acquisition (DDA) mode.

The resulting spectra from each run were searched separately against the 733788-X101SC20124467-Z02-Chenopodium quinoa Willd.-NCBI.fasta database by Proteome Discoverer 2.4 (PD 2.4, Thermo). Peptide spectrum matches (PSMs), with credibility of more than 99%, were identified, and the identified PSMs and proteins were retained and performed with FDR no more than 1.0%. Proteins with fold changes in a comparison >1.2 or <0.83 and unadjusted significance level *p* < 0.05 were considered as DEPs. The DEPs were then analyzed by GO and KEGG enrichment analyses. The mass spectrometry proteomics data were deposited to the ProteomeXchange Consortium (http://proteomecentral.proteomexchange.org (accessed on 24 May 2021)) via the iProX partner repository [34] with the dataset identifier PXD026210.

### 2.4. Correlation Analysis between Proteomic and Transcriptomic Results

The DEGs and the DEPs were separately counted, and the Venn diagrams were plotted according to the counted results. Correlation analysis was performed for the differential multiples of DEGs or DEPs identified in both transcriptomic analysis and proteomic analysis by R (version 3.5.1).

### 2.5. Quantitative Real Time PCR (qRT-PCR) Analysis

In this research, 12 DEGs were selected randomly for qRT-PCR verification, and the coding sequence (CDS) of these selected 12 DEGs are listed in the Appendix A. *CqACTIN* was used as the endogenous control. The primers were designed using Primer Premier 5.0 (Premier) and are listed in Appendix A. Total RNA was extracted by the CTAB method and subjected to DNase treatment (Takara, Japan). The first-strand cDNA was synthesized using M-MLV reverse transcriptase (Takara, Japan) with oligo d(T)18 primer. The qRT-PCR program contains a preliminary step of 2 min at 50 °C, 10 min at 95 °C, followed by 40 cycles of 95 °C for 60 s, 56 °C for 20 s, and 72 °C for 15 s. Three independent biological replicates and three technical replicates were used. The primer efficiency was tested by generating standard curves, and the data were analyzed by the comparative ΔΔCT method.

### 2.6. Physiological Indexes Detection

In this research, 4-week-old quinoa seedlings of ‘NL-6′ treated with water, 300 mM NaCl, or 300 mM NaCl with 100 μM ACC for 2–3 d were used for examination of physiological indexes. The nitrogen content and relative level of total chlorophyll were measured by PJ-4N plant nutrition analyzer, and the relative permeability of cell membrane, damage rate of leaves, malondialdehyde (MDA) content, soluble sugar level, and SOD activity were analyzed as previously described [35]. Three independent biological replicates and three technical replicates were used in the experiments.

### 2.7. Statistical Analyses

Statistical analyses were performed by SAS, and the statistical significance of the difference was evaluated by ANOVA. Means followed by the same letter were not significantly different at the α = 0.05 level.

## 3. Results

### 3.1. Gene Identification and DEGs Analysis in Transcriptome

To investigate ethylene-regulated salt responses in quinoa, the 4-week-old H_2_Or, SALTr, and ACCr samples were used for transcriptomic analysis. The principal component analysis (PCA) showed the differences among different treatments and confirmed the reliability of the sequencing results (Appendix A). A total of 60,602 genes were identified, and the genes with corrected *p*-value < 0.05 and absolute fold change ≥2 were considered as significant DEGs. The DEGs between SALTr and H_2_Or were recognized as the components in salt responses of quinoa. The DEGs between SALTr and ACCr were recognized as functioning in ethylene-regulated salt responses of quinoa, and the DEGs between ACCr and H_2_Or were thought to be involved in ethylene responses or salt responses of quinoa. The DEGs in these three comparisons are presented in the Appendix A. The heat maps with hierarchical clustering, which show relative expression of the DEGs in these comparisons, are presented in Appendix A.

### 3.2. DEGs Detection in Ethylene and Salt Responses of Quinoa

In order to confirm the DEGs in ethylene and salt responses in quinoa, multiple comparisons of DEGs among SALTr-vs-H_2_Or, SALTr-vs-ACCr, and ACCr-vs-H_2_Or were conducted, and 637 DEGs were detected in the overlapping region, which was thought to be playing a role in ethylene-regulated salt responses of quinoa (Figure 1A, Appendix A). GO and KEGG enrichment analysis of these 637 DEGs suggested that most DEGs are involved in antioxidant activity, peroxidase activity, cellulose synthase activity, cellular carbohydrate biosynthesis in phenylpropanoid biosynthesis, cutin, suberine and wax biosynthesis, hormone signal transduction, and the MAPK signaling pathway (Figure 1B,C). Annotations of the DEGs including alcohol-forming fatty acyl-CoA reductases (FARs), glucosyltransferases (GTs), glycerol-3-phosphate acyltransferases (GPATs), chalcone synthases (CHSs), beta-glucosidases (BGLUs), beta-galactosidases (BGALs), cellulose synthases (CESs), trichome birefringence-like proteins (TBLs), cytochrome P450s (CYPs), glutathione S-transferases (GSTs), lipoxygenases (LOXs), pathogenesis-related proteins (PRs), GDSL esterase/lipases (GELPs), and ATP-binding cassette (ABC) transporters are listed in Table 2 and Appendix A. The heat map, which shows relative expression of the DEGs in ethylene and salt responses of quinoa, is presented in Appendix A.

### 3.3. Protein Identification and DEPs Analysis

The H_2_Op, SALTp, and ACCp samples were analyzed by proteomics. The PCA analysis showed the differences among different treatments and confirmed the reliability of the proteomic analysis results (Appendix A). A total of 9672 proteins were identified, and the proteins with fold change >1.2 or <0.83 and unadjusted significance level *p* < 0.05 were considered as DEPs. Similar to the analysis of DEGs, the DEPs between SALTp and H_2_Op were recognized as the components of quinoa salt responses, and the DEPs between SALTp and ACCp were recognized as functioning in ethylene-regulated salt responses of quinoa, and the DEPs between ACCp and H_2_Op were thought to be involved in ethylene responses or salt responses of quinoa. The DEPs in these three comparisons are presented in Appendix A. The heat maps with hierarchical clustering, which show relative expression of the DEPs in these comparisons, are presented in Appendix A.

### 3.4. DEPs Analysis in Ethylene Regulated Salt Responses

In order to study the DEPs in ethylene-regulated salt responses in quinoa, multiple comparisons of DEPs among SALTp-vs-H_2_Op, SALTp-vs-ACCp, and ACCp-vs-H_2_Op were conducted, and nine DEPs were overlapped in the comparisons and may play roles in ethylene-regulated salt responses of quinoa (Figure 2A, Appendix A). These nine DEPs were annotated as POD, 12-oxophytodienoate reductase (OPR), PR, aquaporin, pyrophosphatase (PPase), Kunitz-type trypsin inhibitor (KTI), bark storage protein (BSP), and O-methyltransferase (OMT), respectively (Table 2, Appendix A). The heat map, which shows relative expression of the DEPs in ethylene-regulated salt responses of quinoa, is presented in Appendix A. GO and KEGG enrichment analysis suggested that the DEPs may mostly function in oxidative stress and diverse metabolic processes in vitamin B6 metabolism, phenylpropanoid biosynthesis, linolenic acid metabolism, plant–pathogen interactions, hormone signal transduction, and the MAPK signaling pathway (Figure 2B,C).

### 3.5. Correlation between the Proteomic and Transcriptomic Results

Correlation analysis between the transcriptomic data and the proteomic result identified 184 genes/proteins, which were differentially expressed when treated with salt in quinoa (Figure 3, Appendix A). A total of 189 genes/proteins were detected in salt responses or ethylene responses of quinoa (Figure 3, Appendix A). Among them, 17 genes/proteins may function in quinoa responses to ethylene and salt stress (Figure 3, Appendix A).

The correlation analysis between the transcriptome and proteome also detected 117, 113, and 69 proteins differentially expressed in the comparisons between SALT and H_2_O samples, between ACC and H_2_O samples, and between ACC and SALT samples, respectively, but no expression difference was detected in their transcript levels, suggesting a possible presence of post-transcriptional modification in the proteins (Figure 3, Appendix A).

In contrast, it was found that 3934, 2791, and 804 genes were detected differentially expressed at the transcript level but not at the protein level in the comparisons between SALT and H_2_O samples, between ACC and H_2_O samples, and between ACC and SALT samples, respectively (Figure 3, Appendix A), suggesting that stress-regulated molecules are more likely altered at the transcript level when challenged.

### 3.6. Verification of RNA-seq Results by qRT-PCR

In order to verify the results obtained from the quinoa transcriptomic and proteomic analysis in ethylene-regulated salt responses, 12 DEGs were randomly selected, and their relative expression levels were examined in the quinoa seedlings treated with water, NaCl, and NaCl with ACC for 0, 3, 6, 9, 12, 24, and 36 h. The expressions of the reference genes under the different treatments are shown in Appendix A. The results showed that the expressions of *CqNRT2.1* and *CqACO1* were increased to a peak after 6 h of salt treatment and then began to decrease, while the expressions of *CqCSI*, *CqPER12*, *CqFK*, and *CqPDP* were increased to a peak after 12 h of salt treatment and then began to decrease (Figure 4). The expression of *CqABCB* kept decreasing in SALT and ACC samples (Figure 4). Together, the expressions of these 12 DEGs were obviously affected by salt and ethylene, suggesting that they may play important roles i.

### 3.7. Physiological Alterations by Ethylene and Salt Stress

In order to examine the physiological changes in the H_2_O-, SALT-, and ACC-treated samples (Figure 5A), the nitrogen content, SPAD value, relative permeability of cell membrane, damage rate of leaves, MDA content, soluble sugar level, and SOD activity were detected in these samples. The results indicated that salt treatment rendered higher relative permeability of cell membranes, damage rate of leaves, MDA, and soluble sugar levels, while ethylene treatment reduced them (Figure 5B–E). The SOD activity was activated by salt treatment, which was enhanced by ethylene (Figure 5F). However, the relative content of total chlorophyll denoted by the SPAD value, and the N content were reduced due to salt treatment (Figure 5G). The effects of salt on the SPAD value and N content were alleviated by ethylene treatment, although their contents in the ACC sample were still lower than in the untreated sample (Figure 5G). Taken together, it was concluded that ethylene may regulate salt responses in different ways in quinoa.

## 4. Discussion

Quinoa, an ancient crop native to South America, has high nutritional value and health-promoting phytochemicals in seeds and has received increasing world-wide attention in the past decade [8,9,36]. Quinoa is resistant to multiple abiotic stresses including drought, cold, and salinity [9,10]. Salt stress is a major abiotic stress and affects ~6.5% of the total land of the world [9]. The effects of salt stress on plants are mainly divided into two components, the nonspecific osmotic stress that causes water deficit, and the specific ion effects that provoke the accumulation of toxic ions. Quinoa plants shows significant salt tolerance, but the research on quinoa responses to salt stress is still limited. High-throughput transcriptomic analysis provides a way to excavate molecular mechanism of the quinoa salt responses at the genomic scale [10,21,30,31]. Considering the importance of differentially expressed proteins in most biological processes, examination of the protein level change is more valuable and practical. Unfortunately, little is known about the molecular regulation of quinoa at the proteomic scale. In this research, 4-week-old quinoa seedlings treated with water, NaCl, and NaCl with ACC were analyzed by transcriptional sequencing and proteomics. The identified DEGs and DEPs were analyzed by GO and KEGG, and their correlation analyses was conducted. The study provides a strong foundation for further research on the molecular regulation of quinoa to ethylene and salt stress.

### 4.1. Plant Hormones Play Regulatory Roles in Quinoa Responses to Ethylene and Salt Stress

Plant hormones play important roles in various stress responses. For instance, ABA is thought to be essential for plant responses to abiotic stresses in many plant species including wheat, rice, and *Magnolia wufengensis* [19,25,37]. For example, it was reported that the rice *Osnced5* mutant reduced the ABA level and decreased salt tolerance, while *OsNCED5* overexpression increased the ABA level and enhanced salt tolerance, indicating the importance of ABA to the salt tolerance of rice [37]. In quinoa, the gene encoding for 9-cis-epoxycarotenoid dioxygenase (NCED) in ABA biosynthesis was strongly induced after salt treatment [30,38]. Several PP2Cs in ABA signaling were detected highly up-regulated in quinoa under salt stress by transcriptional sequencing [21]. In the present study, only one abscisic acid receptor, PYL4 (LOC110710755), was detected functioning in ethylene-regulated salt responses, one abscisic acid receptor PYL4 (LOC110714604) detected playing roles in non-ethylene-regulated salt responses, and one abscisic acid receptor PYL4 (LOC110715607) detected playing roles in ethylene responses but not in salt tolerance of quinoa (Table 2, Appendix A), suggesting that crosstalk between ABA and ethylene may exist in the quinoa stress responses.

Other hormones were also reported to be involved in quinoa stress responses. TIFY 10A, a JA response repressor, was strongly induced in responses to salt, while the GA 3-betadioxygenase (GA3OX4) in gibberellin biosynthesis was strongly inhibited under salt treatment [21]. In this study, three jasmonate-induced proteins (JIPs; LOC110715081, LOC110711071, and LOC110733576) were detected. In addition, it was reported that one ethylene receptor, ETR1, and one ethylene responsive factor (ERF) were strongly induced by salt treatment in quinoa [21,30]. In the present study, three ERFs (LOC110729845, LOC110730331, and LOC110719638) were detected in ethylene-regulated salt responses of quinoa (Table 2, Appendix A). These findings broaden our understanding of the phytohormone-mediated regulations in the quinoa stress responses.

In the present study, we also found the other novel genes/proteins responding to salt and ethylene in quinoa. For instance, one auxin response factor (ARF; LOC110719716), one auxin binding protein (ABP; LOC110715799), and one cytokinin riboside 5’-monophosphate phosphoribohydrolase (LOG; LOC110738584) were detected (Table 2, Appendix A). ARF and ABP function in the auxin signaling pathway, and LOG functions in the release of a ribose 5’-monophosphate from a cytokinin nucleotide to form a biologically active cytokinin [39]. Our results indicated that auxin and cytokinin may play roles in ethylene-regulated salt responses of quinoa, and these genes/proteins may be important for the crosstalk of plant hormones.

The auxin efflux carrier (LOC110691454 and LOC110736434), auxin transporter (LOC110706251), ARF (LOC110736906, LOC110715765, and LOC110714183), and ABP (LOC110691560) were detected from this study, and they may be involved in non-ethylene-regulated salt responses (Appendix A). In contrast, the detected auxin response factor (LOC110714183), auxin efflux carrier (LOC110691454 and LOC110736434), auxin transporter (LOC110706251), and ARF (LOC110736906 and LOC110715765) may be involved in ethylene responses but not in salt tolerance of quinoa (Appendix A). Taken together, these results suggest that the plant hormone auxin may play diverse roles in quinoa.

### 4.2. ROS Scavenging Enzymes Function in Quinoa Responses to Ethylene and Salt Stress

Salt stress causes ROS accumulation and oxidative stress aggravation [11]. ROS damage nucleic acids, oxidize proteins, and cause lipid peroxidation, while the antioxidant enzymes including GST, SOD, POD, and CAT neutralize the salt-induced ROS accumulation to protect plants from destructive oxidative reactions [12,17,18]. In detail, SOD dismutates O_2_^−^ into H_2_O_2_, which is decomposed into water and oxygen by CAT in the peroxisomes. POD mainly catalyzes substrate oxidation with H_2_O_2_ as an electron acceptor in vacuoles and cell walls in plants [40,41,42]. In plants, GSTs are multifunctional enzymes existed in different classes (Phi, Tau, Zeta, Theta, and others) and play important roles in cellular detoxification of xenobiotic protection against oxidative stress as well as diverse ligand-binding activities [43].

In this research, 9 GSTs (LOC110724460, LOC110696392, LOC110724461, LOC110728060, LOC110711174, LOC110711174, LOC110739278, LOC110713696, and LOC110727188) and 16 PODs (LOC110682117, LOC110682546, LOC110685850, LOC110692926, LOC110699378, LOC110724764, LOC110735668, LOC110694635, LOC110735670, LOC110681844, LOC110687369, LOC110690635, LOC110727528, LOC110699380, LOC110684661, and LOC110704239) were detected in the quinoa responses to ethylene and salt stress (Table 2, Appendix A). The ROS scavenging enzymes, POD5, had been reported to be functioning in salt responses of quinoa by RNA-seq [21]. In this study, two POD5 including LOC110692926 and LOC110727528 were detected in ethylene-regulated salt responses of quinoa. On the other hand, 16 PODs including LOC110683143, LOC110729735, and LOC110699379 may play roles in non-ethylene-regulated salt responses (Appendix A), and 23 PODs including LOC110685846, LOC110704240, LOC110707569, LOC110711884, and LOC110704238 are probably involved in ethylene responses but not in salt responses in quinoa (Appendix A). The PODs were also detected as core salt-responsive genes in both salt-tolerant quinoa and salt-sensitive quinoa by a previous RNA-seq research [21]. In this study, the SOD activity was also detected and activated by salt treatment in quinoa (Figure 5F). In addition, it was shown that ethylene enhances the SOD activity in salt responses of quinoa (Figure 5F), providing evidence that ethylene may mediate ROS to regulate salt tolerance of quinoa.

### 4.3. Osmotic Adjustment Is Important for Quinoa Responses to Ethylene and Salt Stress

High concentrations of NaCl in salt stress generate K^+^ and H^+^ fluxes in quinoa roots to the apoplast, so leaf osmoregulation, K^+^ retention, Na^+^ exclusion, and ion homeostasis confer quinoa salt tolerance [44]. In addition to these inorganic ions, the accumulation of organic substances including protein, sugars, proline, and total phenolics is also attributed to improve the quinoa salt tolerance [15,16]. The high affinity K^+^ transporters (HKT1.2) play a key role in Na^+^ load into bladder cells in quinoa, and the Na^+^ in bladder cells is then collected in the bladder hairs and washed off by rain [45]. Quinoa plants were reported to accumulate more Na^+^ than K^+^ under salinity stress, because the K^+^/Na^+^ ratio was detected to be decreased with increasing of salt concentration [44]. The cell anion channel (SLAH), nitrate transporter (NRT), and chloride channel protein (C1C) were also activated by salt stress, indicating their possible functions in salt responses of quinoa [10].

In this research, one HKT (LOC110710365), four phosphate transporters (PTs) (LOC110689438, LOC110720352, LOC110689401, and LOC110717783), 1 Na^+^/metabolite cotransporter (SMT) (LOC110727554), four NRTs (LOC110688100, LOC110684366, LOC110715529, and LOC110684367), one cation/H^+^ antiporter (CAH) (LOC110699138), one Na^+^/Ca^2+^ exchanger (NCL) (LOC110709231), and one aquaporin (LOC110697673) were detected in the quinoa responses to ethylene and salt stress (Table 2, Appendix A). However, three Na^+^/H^+^ exchangers (LOC110702071, LOC110692001 and LOC110737010), one H^+^/Ca^2+^ exchanger (LOC110738999), one CAH (LOC110682708), and two cation/Ca^2+^ exchangers (LOC110688074 and LOC110708567) were detected only in salt responses but not in ethylene responses, indicating these genes/proteins may function in non-ethylene-regulated salt responses (Appendix A). No iron transporter genes/proteins were detected in the ethylene responses of quinoa (Appendix A). These results provide useful information for possible assimilation and transportation of the inorganic ions in quinoa stress responses.

In addition to inorganic ions, the accumulation of organic solutes, including soluble sugars and proline, also decreases osmotic potential under salt stress [46]. In quinoa, the total sugar is increased due to salt treatment [47]. In this research, one organic cation/carnitine transporter (OCTN) (LOC110737811), one aluminum-activated malate transporter (ALMT) (LOC110688161), three bidirectional sugar transporters (SWEETs) (LOC110725786, LOC110735791, and LOC110732264), one polyol transporter (POT) (LOC110722677), one amino acid transporter (AAT) (LOC110733528), one nucleobase-ascorbate transporter (NAT) (LOC110708068), six ABC transporters (LOC110712440, LOC110707705, LOC110729523, LOC110695413, LOC110721597, and LOC110722212), six glucan endonucleases (GLUs) (LOC110717180, LOC110717177, LOC110699037, LOC110717159, LOC110699174, and LOC110736258), and two sucrose synthases (SSs) (LOC110727927 and LOC110689796) were detected in the quinoa responses to ethylene and salt stress (Table 2, Appendix A). Although the genes encoding GLUs and SSs in carbohydrate metabolic process were previously detected in the salt responses of quinoa [21], the functions of the OCTN, ALMT, SWEETs, POT, AAT, NAT, and ABC transporters in salt responses of quinoa had not been reported in quinoa. The regulation mechanisms definitely need to be explored in future studies.

### 4.4. Cell Wall Structural Proteins Respond to Ethylene and Salt Stress in Quinoa

The levels of principal structural component of the plant cell wall such as lignins, pectins, celluloses, and hemicelluloses are affected by salt stress, which induce the alteration of cell wall elasticity [48,49]. Previously, it was reported that transcriptional changes of the genes involved in cell wall organization could been detected by RNA-seq after salt treatment of quinoa seedlings [30]. The genes involved in suberin and cutin biosynthesis, photosynthesis, and chloroplast were also reported to be significantly changed due to salt treatment in the bladder cells of quinoa [10]. TBLs encode the cell wall polysaccharide specific O-acetyltransferases and are probably involved in maintaining esterification of pectins [50]. In *Arabidopsis*, the functional study of the cellulose synthesis in salt tolerance had been previously reported [51]. In this research, 2 TBLs (LOC110715157 and LOC110685228) were detected differentially expressed (Table 2, Appendix A), and 5 CESs (LOC110715976, LOC110717430, LOC110689768, LOC110689717, and LOC110721870) in cellulose synthesis were detected. In addition, two BGLUs (LOC110739769 and LOC110724275), two BGALs (LOC110682558 and LOC110685863), and four glycine-rich cell wall structural proteins (GRPs) (LOC110732550, LOC110730178, LOC110730179, and LOC110732549), which may be involved in cell wall structure and elasticity in quinoa, were detected (Table 2, Appendix A). All these findings strongly support the importance of cell wall structure and elasticity in the quinoa stress responses.

### 4.5. Secondary Metabolism-Associated Proteins Respond to Ethylene and Salt Stress in Quinoa

Betalain is a tyrosine-derived, red–violet, and yellow pigment in quinoa with antioxidant activity, which plays important roles in salt responses [52]. For example, CqCYP76AD1-1 was reported in the betalain biosynthesis process in quinoa [53,54]. In this research, one CqCYP76AD1 (LOC110731693) was detected in the ethylene-regulated salt responses, although its molecular mechanism in the responses is unclear. The methyltransferases (MTs), GTs, and GPATs are transferases that transfer methyl, glucosyl, and acyl groups from one compound to another, respectively. The CHSs condense a phenylpropanoid CoA ester with three acetate units from malonyl-CoA molecules and cyclize the resulting intermediate to produce a chalcone, which is the precursor of diverse flavonoids [55]. The GELPs have high potential to be used in the hydrolysis and synthesis of important ester compounds [56]. It was reported that ectopic expression of *Arabidopsis* glycosyltransferase *UGT85A5* enhances salt tolerance in tobacco, but knock down of the corresponding genes decreases salt tolerance at seedling and reproductive stages of rice [57,58].

In this study, 10 GTs (LOC110714725, LOC110729660, LOC110706607, LOC110739778, LOC110683464, LOC110722666, LOC110711362, LOC110738265, LOC110735480, and LOC110718641), 4 GPATs (LOC110691783, LOC110722317, LOC110733316, and LOC110714505), 3 CHSs (LOC110691992, LOC110691988, and LOC110702060), 5 GELPs (LOC110735138, LOC110712448, LOC110709557, LOC110717860, and LOC110703315), 6 CYPs (LOC110731693, LOC110739776, LOC110718248, LOC110727125, LOC110681912, and LOC110724693), and 2 MTs (LOC110703261 and LOC110728006) were detected in the quinoa responses to ethylene and salt stress (Table 2, Appendix A), suggesting that these gene/protein-mediated diverse metabolisms may be involved in the quinoa ethylene and salt responses.

In addition, 33 CYPs including LOC110711004, LOC110684386, LOC110707034, LOC110711698, LOC110715344, and LOC110732720 were detected only in salt responses, suggesting their possible functions in salt responses but not ethylene responses (Appendix A). In contrast, one CHS (LOC110691988) and nine GELPs (LOC110700001, LOC110719694, LOC110728839, LOC110731812, LOC110693712, LOC110730528, LOC110700478, LOC110695766, and LOC110709613) were activated by ethylene but not salt stress (Appendix A). All these results indicate a complication of the molecular regulations by secondary metabolism-associated proteins in quinoa.

### 4.6. Early Response Genes and Late Response Genes in Quinoa

It was reported that many genes are divided into two categories, namely early response genes and late response genes, depending on their different activation patterns in response to stimuli. The early response genes, which are also called primary response genes, are induced without de novo protein synthesis, while the late response genes, which are also called secondary response genes, require de novo protein synthesis and are induced more slowly because that synthesis needs signaling molecules or cytokines [59,60].

In this research, the correlation between proteome and transcriptome was analyzed, and the results are shown in Figure 3, Table 2, and Appendix A. For example, the genes/proteins (GST (LOC110739278) and GLU1 (LOC110717177)) differentially expressed in both transcript and protein levels, belong to early response genes, and their proteins had already been synthesized and could be detected at early times.

The genes/proteins that were differentially expressed in transcript levels but not protein levels belonged to late response genes. Their protein levels did not accumulate within 24 h of treatment. It was suggested that their protein levels could be changed in later hours. In this research, the genes/proteins including JIPs (LOC110715081, LOC110711071, and LOC110733576), GSTs (LOC110724460, LOC110696392, LOC110724461, LOC110728060, and LOC110711174), and PODs (LOC110682546, LOC110685850, LOC110692926, LOC110699378, LOC110724764, LOC110735668, LOC110694635, LOC110735670, LOC110681844, LOC110687369, LOC110690635, LOC110727528, LOC110699380, and LOC110684661) were differentially expressed in transcript levels but not in protein levels, suggesting that these genes may belong to the late response genes.

The genes/proteins including GSTs (LOC110713696 and LOC110727188), POD64 (LOC110682117), POD12 (LOC110704239), PIP2-5 (LOC110697673), GLU1 (LOC110717180), SSs (LOC110727927 and LOC110689796), GT (LOC110718641), GELP (LOC110703315), and MTs (LOC110703261 and LOC110728006) were differentially expressed in protein levels but not in transcript levels, suggesting that post-transcriptional modifications may occur in the genes/proteins.

## 5. Conclusions

The proposed molecular mechanism of ethylene-regulated salt responses in quinoa is complex. Under salt stress, ROS scavenging enzymes including GSTs and PODs; transporters and solutes in osmotic adjustment including HKT, PT, Na^+^/metabolite cotransporter, high-affinity Na^+^ transporters, cation/H^+^ antiporter, Na^+^/Ca^2+^ exchanger, aquaporin, bidirectional sugar transporters, polyol transporter, and sucrose synthases; cell wall structural proteins including GLCs, β-GALs, CESs, TBLs, and GRPs; and secondary metabolism-associated proteins including GTs, GPATs, CHSs, GELPs, CYPs, and MTs are activated in responses to ethylene and salt stress in quinoa. Plant hormones, including AUX, ABA, JA, and CK, also play important roles in the responses. Considering the large number of transporters in osmotic adjustment identified in the ethylene-regulated salt responses in quinoa, it is concluded that osmotic adjustment is probably one of the main regulations for quinoa when challenged by salt stress.

## Figures and Tables

**Figure 1 plants-10-02281-f001:**
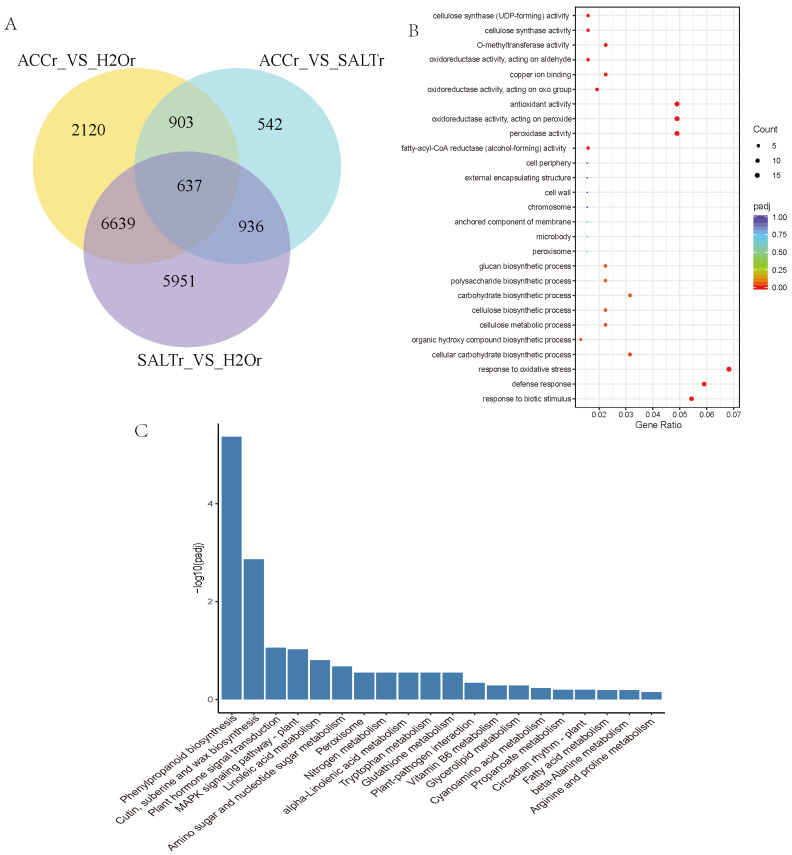
Multiple comparison of H_2_Or-vs-SALTr, H_2_Or-vs-ACCr, and SALTr-vs-ACCr. (**A**) Venn diagrams of DEGs in multiple comparisons. (**B**) GO enriched scatter plot of DEGs in ethylene and salt responses of quinoa. Rich factor refers to the ratio of the gene number enriched in the pathway to the number of annotated genes. The bigger the Rich factor, the more significant the enrichment is. The padj value is the corrected *p*-value after multiple hypotheses testing, which ranges from 0 to 1. The closer to zero, the more significant the enrichment is. (**C**) The top 20 KEGG enrichment analyses of DEGs in ethylene and salt responses of quinoa. The *y*-axis indicates the number of DEGs, and the *x*-axis shows the processes/components in different functions.

**Figure 2 plants-10-02281-f002:**
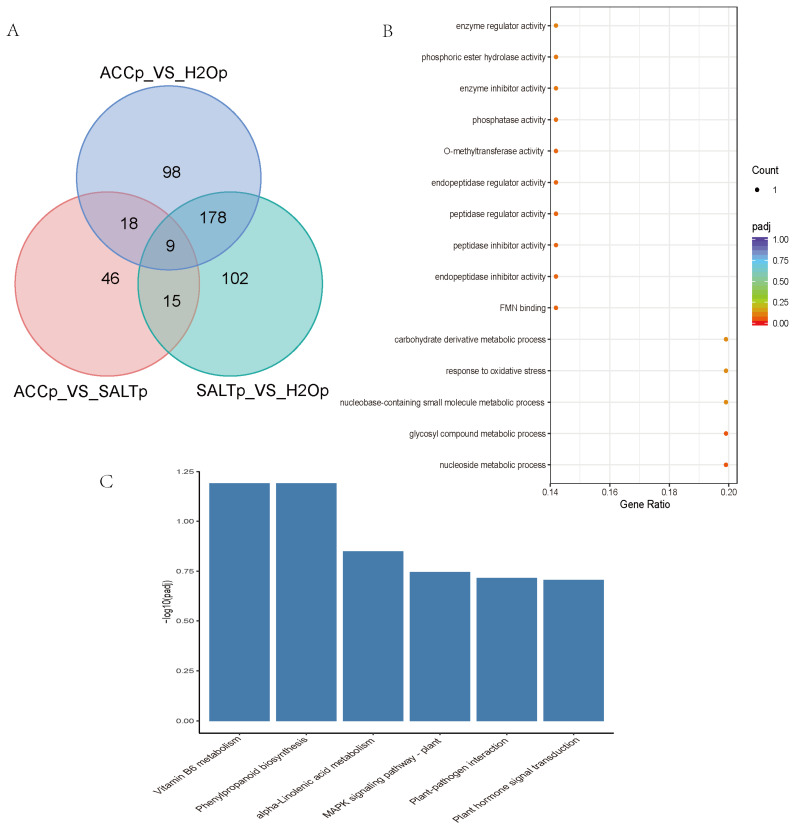
Multiple comparison of H_2_Op-vs-SALTp, H_2_Op-vs-ACCp, and SALTp-vs-ACCp. (**A**) Venn diagrams of DEPs in multiple comparison. (**B**) GO enriched scatter plot of DEPs in ethylene and salt responses of quinoa. Rich factor refers to the ratio of the gene number enriched in the pathway to the number of annotated genes. The bigger the Rich factor, the more significant the enrichment is. The padj value is the corrected *p*-value after multiple hypotheses testing, which ranges from 0 to 1. The closer to zero, the more significant the enrichment is. (**C**) The top 20 KEGG enrichment analysis of DEPs in ethylene and salt responses of quinoa. The Y-axis indicates the number of DEPs, and the X-axis shows the processes/components in different functions.

**Figure 3 plants-10-02281-f003:**
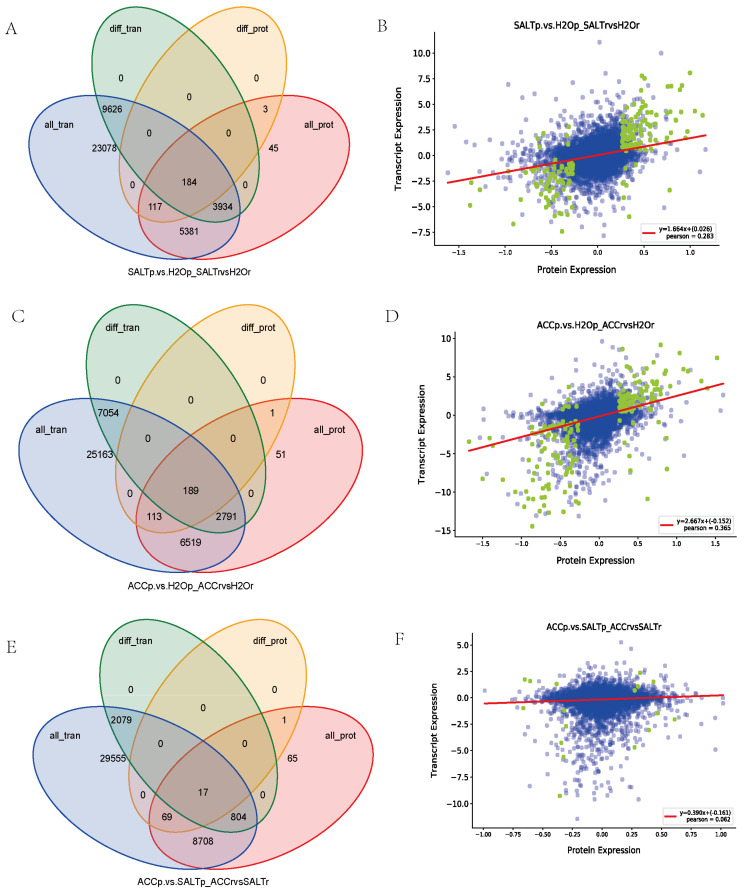
The correlation analysis between proteome and transcriptome by Venn diagrams and scatter plot of expression correlation. (**A**,**C**,**E**) tVenn diagrams of genes, proteins, DEGs, and DEPs between SALT and H_2_O samples (**A**), between ACC and H_2_O samples (**C**), and between ACC and SALT samples (**E**), respectively; all_tran represents all the genes obtained from the transcriptome, diff_tran represents the DEGs identified by transcriptome, all_prot represents all the proteins identified by proteome, and diff_prot represents the DEPs identified by proteome. (**B**,**D**,**F**) Scatter plots of expression correlation between SALT and H_2_O samples (**B**) between ACC and H_2_O samples (**D**) and between ACC and SALT samples (**F**), respectively. The abscissa is the differential multiple of proteins, the ordinate is the differential multiple of corresponding genes, and the correlation coefficient and *p* value of the transcriptome and proteome are also shown in the figures. The points represent proteins/genes: the blue points represent non-differential proteins/genes, and the green points represent DEPs/DEGs.

**Figure 4 plants-10-02281-f004:**
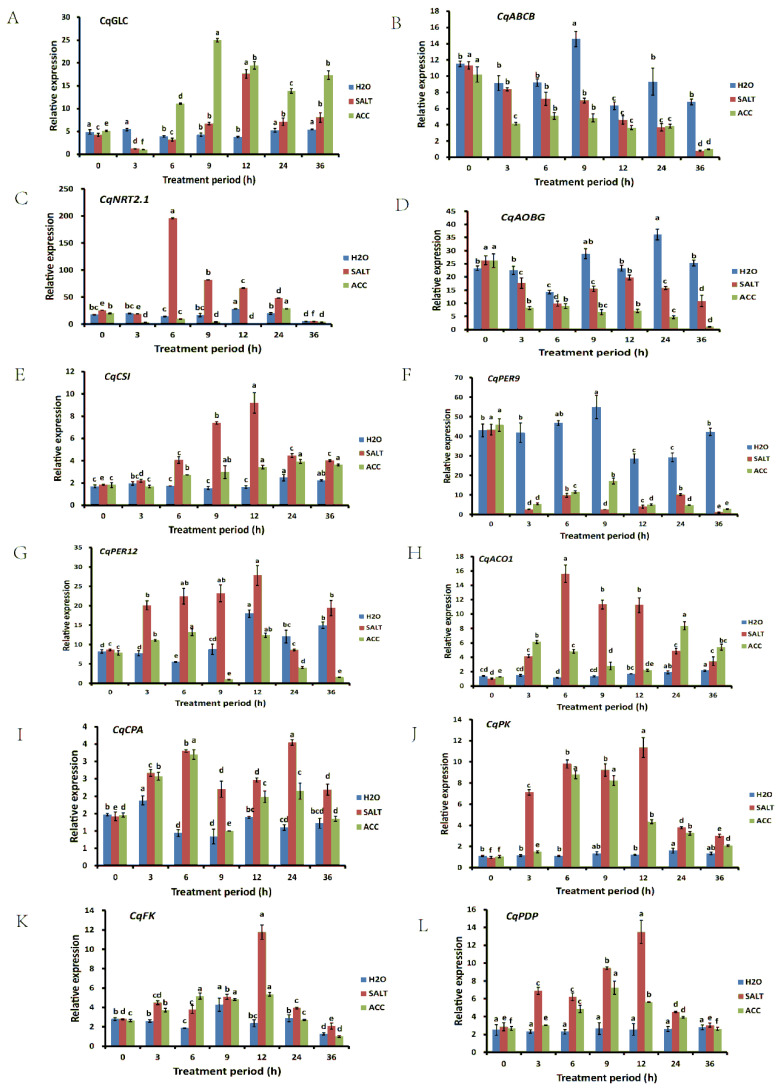
Gene expression verification by qRT-PCR. The expressions of *CqGLC* (**A**), *CqABCB* (**B**), *CqNRT2.1* (**C**), *CqAOBG* (**D**), *CqCSI* (**E**), *CqPER9* (**F**), *CqPER12* (**G**), *CqACO1* (**H**), *CqCPA* (**I**), *CqPK* (**J**), *CqFK* (**K**), and *CqPDP* (**L**) were detected under different treatments with different treatment periods. The differences between samples at different treatment periods were analyzed, and the statistical significance of the difference was confirmed by ANOVA at α = 0.05 level.

**Figure 5 plants-10-02281-f005:**
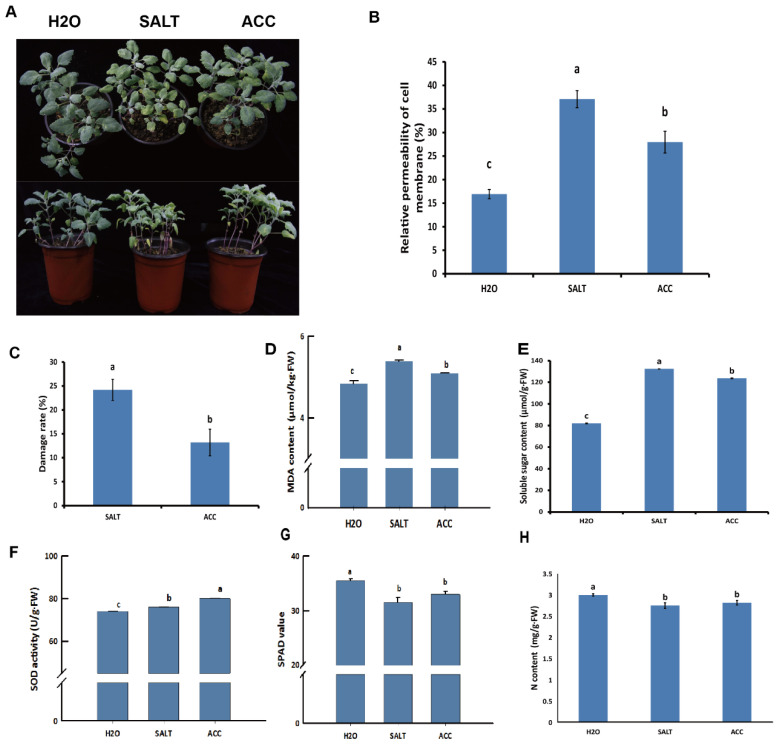
Detection of physiological changes of quinoa seedlings with different treatments. The treated quinoa seedlings were photographed (**A**), and the relative permeability of cell membrane (**B**), damage rate of leaves (**C**), MDA content (**D**), soluble sugar level (**E**), SOD activity (**F**), SPAD value (**G**), and nitrogen level (**H**) of quinoa seedlings were detected. The statistical significance of the difference was analyzed by ANOVA at α = 0.05 level.

**Table 1 plants-10-02281-t001:** The abbreviations of quinoa samples used in this research.

Abbreviations	Detailed Information	Omics Used in
H_2_Op	Water-treated seedlings	proteomics
SALTp	Salt-treated seedlings	proteomics
ACCp	Salt- and ACC-treated seedlings	proteomics
H_2_Or	Water-treated seedlings	transcriptomics
SALTr	Salt-treated seedlings	transcriptomics
ACCr	Salt- and ACC-treated seedlings	transcriptomics

**Table 2 plants-10-02281-t002:** A summary of candidate proteins/genes in ethylene and salt responses of quinoa.

Gene ID	Symbol	Description	Function Category	Classification
LOC110729845	ERF	ethylene-responsive transcription factor	plant hormone signaling	late response gene
LOC110730331	ERF104	ethylene-responsive transcription factor 104	plant hormone signaling	late response gene
LOC110719638	ERF2	ethylene-responsive transcription factor 2	plant hormone signaling	late response gene
LOC110719716	ARF9	auxin response factor 9	plant hormone signaling	late response gene
LOC110715799	ABP19a	auxin-binding protein 19a	plant hormone signaling	late response gene
LOC110710755	PYL4	abscisic acid receptor PYL4	plant hormone signaling	late response gene
LOC110715081	JIP	jasmonate-induced protein	plant hormone signaling	late response gene
LOC110711071	JIP	jasmonate-induced protein	plant hormone signaling	late response gene
LOC110733576	JIP	jasmonate-induced protein	plant hormone signaling	late response gene
LOC110738584	LOG5	cytokinin riboside 5′-monophosphate phosphoribohydrolase	plant hormone biosynthesis	late response gene
LOC110724460	GST	glutathione S-transferase	ROS scavenging	late response gene
LOC110696392	GST	glutathione S-transferase	ROS scavenging	late response gene
LOC110724461	GST	glutathione S-transferase	ROS scavenging	late response gene
LOC110728060	GST	glutathione S-transferase	ROS scavenging	late response gene
LOC110711174	GST3	microsomal glutathione S-transferase 3	ROS scavenging	late response gene
LOC110739278	GST	glutathione S-transferase	ROS scavenging	early response gene
LOC110713696	GST	glutathione S-transferase	ROS scavenging	post-transcriptional modification
LOC110727188	GST	glutathione S-transferase	ROS scavenging	post-transcriptional modification
LOC110682117	POD64	peroxidase 64	ROS scavenging	post-transcriptional modification
LOC110682546	POD72	peroxidase 72	ROS scavenging	late response gene
LOC110685850	POD72	peroxidase 72	ROS scavenging	late response gene
LOC110692926	POD5	peroxidase 5	ROS scavenging	late response gene
LOC110699378	POD4	peroxidase 4	ROS scavenging	late response gene
LOC110724764	POD9	peroxidase 9	ROS scavenging	late response gene
LOC110735668	POD12	peroxidase 12	ROS scavenging	late response gene
LOC110694635	POD55	peroxidase 55	ROS scavenging	late response gene
LOC110735670	POD12	peroxidase 12	ROS scavenging	late response gene
LOC110681844	POD55	peroxidase 55	ROS scavenging	late response gene
LOC110687369	POD42	peroxidase 42	ROS scavenging	late response gene
LOC110690635	POD1	cationic peroxidase 1	ROS scavenging	late response gene
LOC110727528	POD5	peroxidase 5	ROS scavenging	late response gene
LOC110699380	POD4	peroxidase 4	ROS scavenging	late response gene
LOC110684661	POD1	cationic peroxidase 1	ROS scavenging	late response gene
LOC110704239	POD12	peroxidase 12	ROS scavenging	post-transcriptional modification
LOC110710365	HKT5	potassium transporter 5	osmotic adjustment	late response gene
LOC110689438	PT1-3	phosphate transporter 1-3	osmotic adjustment	late response gene
LOC110720352	PHO1	phosphate transporter PHO1	osmotic adjustment	late response gene
LOC110689401	PHO1	phosphate transporter PHO1	osmotic adjustment	late response gene
LOC110717783	PT1-3	phosphate transporter 1-3	osmotic adjustment	late response gene
LOC110727554	SMT	sodium/metabolite cotransporter	osmotic adjustment	late response gene
LOC110688100	NRT2.1	high-affinity nitrate transporter 2.1	osmotic adjustment	late response gene
LOC110684366	NRT2.1	high-affinity nitrate transporter 2.1	osmotic adjustment	late response gene
LOC110715529	NRT3.2	high-affinity nitrate transporter 3.2	osmotic adjustment	late response gene
LOC110684367	NRT2.4	high-affinity nitrate transporter 2.4	osmotic adjustment	late response gene
LOC110699138	CAH20	cation/H^+^ antiporter 20	osmotic adjustment	late response gene
LOC110709231	NCL	sodium/calcium exchanger NCL	osmotic adjustment	late response gene
LOC110697673	PIP2-5	aquaporin PIP2-5	osmotic adjustment	post-transcriptional modification
LOC110737811	OCTN7	organic cation/carnitine transporter 7	osmotic adjustment	late response gene
LOC110688161	ALMT2	aluminum-activated malate transporter 2	osmotic adjustment	late response gene
LOC110725786	SWEET1	bidirectional sugar transporter SWEET1	osmotic adjustment	late response gene
LOC110735791	SWEET1	bidirectional sugar transporter SWEET1	osmotic adjustment	late response gene
LOC110732264	SWEET7	bidirectional sugar transporter SWEET7	osmotic adjustment	late response gene
LOC110722677	POT	polyol transporter	osmotic adjustment	late response gene
LOC110733528	AAT	amino acid transporter	osmotic adjustment	late response gene
LOC110708068	NAT7	nucleobase-ascorbate transporter 7	osmotic adjustment	late response gene
LOC110712440	ABCC15	ABC transporter C family member 15	osmotic adjustment	late response gene
LOC110707705	ABCB8	ABC transporter B family member 8	osmotic adjustment	late response gene
LOC110729523	ABCC10	ABC transporter C family member 10	osmotic adjustment	late response gene
LOC110695413	ABCA2	ABC transporter A family member 2	osmotic adjustment	late response gene
LOC110721597	ABCC15	ABC transporter C family member 15	osmotic adjustment	late response gene
LOC110722212	ABCB25	ABC transporter B family member 25	osmotic adjustment	late response gene
LOC110717180	GLU1	glucan endonucleases-1	osmotic adjustment	post-transcriptional modification
LOC110717177	GLU1	lucan endonucleases-1	osmotic adjustment	early response gene
LOC110699037	GLU1	glucan endonucleases-1	osmotic adjustment	late response gene
LOC110717159	GLU1	glucan endonucleases-1	osmotic adjustment	late response gene
LOC110699174	GLU1	glucan endonucleases-1	osmotic adjustment	late response gene
LOC110736258	GLU1	glucan endonucleases-1	osmotic adjustment	late response gene
LOC110727927	SS	sucrose synthase	osmotic adjustment	post-transcriptional modification
LOC110689796	SS	sucrose synthase	osmotic adjustment	post-transcriptional modification
LOC110739769	BGLU13	beta-glucosidase 13	cell wall construction	late response gene
LOC110724275	BGLU12	beta-glucosidase 12	cell wall construction	late response gene
LOC110682558	BGAL3	beta-galactosidase 3	cell wall construction	late response gene
LOC110685863	BGAL3	beta-galactosidase 3	cell wall construction	late response gene
LOC110715976	CESA	cellulose synthase A	cell wall construction	late response gene
LOC110717430	CESG2	cellulose synthase G2	cell wall construction	late response gene
LOC110689768	CESD5	cellulose synthase D5	cell wall construction	late response gene
LOC110689717	CESA	cellulose synthase A	cell wall construction	late response gene
LOC110721870	CESA	cellulose synthase A	cell wall construction	late response gene
LOC110715157	TBL38	trichome birefringence-like protein 38	cell wall construction	late response gene
LOC110685228	TBL39	trichome birefringence-like protein 39	cell wall construction	late response gene
LOC110732550	GRP1.8	glycine-rich cell wall structural protein 1.8	cell wall construction	late response gene
LOC110730178	GRP1.8	glycine-rich cell wall structural protein 1.8	cell wall construction	late response gene
LOC110730179	GRP1.8	glycine-rich cell wall structural protein 1.8	cell wall construction	late response gene
LOC110732549	GRP1.8	glycine-rich cell wall structural protein 1.8	cell wall construction	late response gene
LOC110714725	CGT	crocetin glucosyltransferase	secondary metabolism	late response gene
LOC110729660	OGT	O-glucosyltransferase	secondary metabolism	late response gene
LOC110706607	7DGT	7-deoxyloganetin glucosyltransferase	secondary metabolism	late response gene
LOC110739778	7DGT	7-deoxyloganetic acid glucosyltransferase	secondary metabolism	late response gene
LOC110683464	OGT	O-glucosyltransferase	secondary metabolism	late response gene
LOC110722666	GT	hydroquinone glucosyltransferase	secondary metabolism	late response gene
LOC110711362	ABGT	anthocyanin 3′-O-beta-glucosyltransferase	secondary metabolism	late response gene
LOC110738265	ABGT	anthocyanin 3′-O-beta-glucosyltransferase	secondary metabolism	late response gene
LOC110735480	UDPGT	UDP-glycosyltransferase 90A1	secondary metabolism	late response gene
LOC110718641	7DGT	7-deoxyloganetin glucosyltransferase	secondary metabolism	post-transcriptional modification
LOC110691783	GPAT	glycerol-3-phosphate acyltransferase	secondary metabolism	late response gene
LOC110722317	GPAT7	glycerol-3-phosphate acyltransferase 7	secondary metabolism	late response gene
LOC110733316	GPAT5	glycerol-3-phosphate acyltransferase 5	secondary metabolism	late response gene
LOC110714505	WSD1	O-acyltransferase WSD1	secondary metabolism	late response gene
LOC110691992	CHS3	chalcone synthase 3	secondary metabolism	late response gene
LOC110691988	CHS3	chalcone synthase 3	secondary metabolism	late response gene
LOC110702060	CHS3	chalcone synthase 3	secondary metabolism	late response gene
LOC110735138	GELP	GDSL esterase/lipase	secondary metabolism	late response gene
LOC110712448	GELP	GDSL esterase/lipase	secondary metabolism	late response gene
LOC110709557	GELP	GDSL esterase/lipase	secondary metabolism	late response gene
LOC110717860	GELP	GDSL esterase/lipase	secondary metabolism	late response gene
LOC110703315	GELP	GDSL esterase/lipase	secondary metabolism	post-transcriptional modification
LOC110731693	CYP76AD1	cytochrome P450 76AD1	secondary metabolism	late response gene
LOC110739776	CYP72A219	cytochrome P450 72A219	secondary metabolism	late response gene
LOC110718248	CYP71A6	cytochrome P450 71A6	secondary metabolism	late response gene
LOC110727125	CYP71A6	cytochrome P450 71A6	secondary metabolism	late response gene
LOC110681912	CYP89A2	cytochrome P450 89A2	secondary metabolism	late response gene
LOC110724693	CYP83B1	cytochrome P450 83B1	secondary metabolism	late response gene
LOC110703261	OMT	O-methyltransferase	secondary metabolism	post-transcriptional modification
LOC110728006	NMT	N-methyltransferase	secondary metabolism	post-transcriptional modification

## Data Availability

The mass spectrometry proteomics data have been deposited to the ProteomeXchange Consortium (http://proteomecentral.proteomexchange.org (accessed on 24 May 2021)) via the iProX partner repository with the dataset identifier PXD026210. The FASTQ files of raw data were uploaded to the NCBI Sequence Read Archive (SRA), and the SRA study accession is PRJNA726352 (accessed on 30 June 2022).

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
