# Peer review of "Genome-Wide Transcriptomic and Proteomic Exploration of Molecular Regulations in Quinoa Responses to Ethylene and Salt Stress"

_plants, 2021, doi:10.3390/plants10112281_

Round 1

Reviewer 1 Report

In this manuscript, following genome-wide transcriptomic and proteomic approaches, the authors studied the molecular mechanism of ethylene-regulated salt stress responses in quinoa, which the United Nations Food and Agricultural Organization (FAO) has declared as a major crop for global food security and sustainability under the current global climate change because of its high level of abiotic stress tolerance. The authors stressed 4-week-old quinoa seedlings using NaCl, and NaCl + ACC, analyzed the stressed plants via transcriptional sequencing and tandem mass tag-based (TMT) quantitative proteomics, and identified nearly 9672 proteins and 60602 genes among which, GST, POD, PT, GLU, BGAL, CES, TBL, GRP, GT, GELP, CYP, and JIP protein genes were significantly differentially regulated. Overall, the study is interesting, methodology is well-designed and results are nicely presented. Some of my concerns are;

Once you use a full name, please use the abbreviations only onwards. For example, use only ACCp (instead of repeating NaCl with ACC).

Some subtopics are better to be rephrased.

Besides, there are many mistakes related to language/grammar throughout the manuscript. I’ll point out some of them below. Please correct them thoroughly in the whole manuscript.

Line 63-64 (climate change)

Line 71 (global warming)

Line 150-151 (24 h what?)

Line 182 Section 2.3 (Rephrase and correct the heading)

Line 221 Section 2.6 (detection NOT detections)

Line 253, 304-305, “a” cannot be used before “multiple comparisons”

Line 386 “has receive”

Line 388 “promising” what?

Line 400 Section 4.1 (Plant hormones “play”)

Line 441 Section 4.3 (adjustment “is” important)

Line 454 “ratio” NOT “ratios”

Line 501 Remove “of a“, before transfer “methyl, …..”

Reviewer 2 Report

Manuscript ID: plants-1403961

Genome-wide transcriptomic and proteomic ex ploration of molecular regulations in quinoa responses to ethylene and salt stress

This manuscript reports the transcriptomic and proteomic changes in quinoa seedlings after treament with salt (300 mM of NaCl) and with salt and ACC (300 mM of NaCl and 0.1 mM ACC) in order to decipher the role of ethylene in quinoa’s salt stress response. Although the transcriptomic and proteomic data reported are impressive, my main concern is related with the experimental design employed to pursuit this goal.

Major comments

Experimental design:

  1. If the goal of this study is to know how the plant hormone ethylene is involved in the quinoa response to salt stress, then it’s more relevant to study the effects of ethylene on specific organs (roots and/or leaves) instead of studying the whole plants, since a lot of information is lost.
  2. NaCl dose. Quinoa is a facultative facultative halophytic plant species and there are different quinoa genotypes that show diferent salt tolerance. It is important to justify the dose of salt used in this study (300 mM) as well as to report clearly the way that quinoa plants were grown. When did the saline treatment start? Did you add to the nutrient solution 300 mM of NaCl directly, or did you gradually increased the NaCl concentration to allow plants to adapt to this high NaCl concentration? Please, clarify this point.
  3. ACC concentration. Please, justify the dose of ACC employed in this study as well as its application to quinoa plants. Did you add ACC together with salt? It is well-known that ethylene plays a key role in the regulation of plant responses to biotic and abiotic stresses but also regulates a wide range of developmental processes (doi: 10.1016/j.tplants.2018.01.003). Thus, in my opinion a treatment with ACC under non-saline conditions is necessary in this study. In addition, as far as the role of ACC is concerned, new evidence suggest that ACC acts as a signaling molecule distinct from its role in ethylene biosynthesis (for review see, https://doi.org/10.3389/fpls.2019.01602).
  4. Quality test. Principal component analysis (PCA) is recommended in order to compare gene expression/proteomic profiles of the three biological samples used in this study.

Quality of qRT-PCR.

The qRT-PCR data do not follow the MIQE guidelines (https://doi.org/10.1373/clinchem.2008.112797). Only a reference gene (CqACTIN) was used in these analysis. It is recommended to show the expression of the reference gene under the different treatments in order to verify its expression in a Supplementary figure.

Data presentation:

The way data are presented does not facilitate the understanding about the role of ethylene on salt stress. It is necessary to show the data obtained in both salt treatment alone and in the combined treatment (salt+ACC). Thus, in my opinion, Fig. 1B should contain the GO enriched scatter plot of DEGs in salt and in double treaments (salt+ACC). Fig. 1C should contain the top 20 KEGG enrichment analysis of DEGs in salt and in double treaments (salt+ACC). Do the same in Fig. 2 with DEPs.

Moreover, a new figure –a heatmap- illustrating the fold changes (log2 treatment/control ratios) of transcripts and proteomic results obtained for the two conditions is strongly recommended. Thus, Table 2 “summary of candidate proteins/genes in ethylene and salt response of quinoa” can be included in supplementary materials

Discussion

A comparison of the effect of salt treatment alone with those combined with ACC (salt+ACC treatment) is lacking. Authors have focused almost on the effect of salt+ACC treatment. Discussion must be re-written highlighting the differences and similarities between salt and salt+ACC treatments.

Other comments:

Literature

Updated review is recommended. For example, in the introduction it is not clear why is important to study the effect on ethylene in quinoa plants challenged with salt. An updated review of the role of ethylene in regulating plant responses to salt stress is, for example doi: 10.3390/biom10060959.

POD

Class III plant peroxidases (EC 1.11.1.7) are mainly secreted into vacuoles and cell walls (see refs bellow). In line 427 it is stated that “POD mainly catalyzes substrate oxidation with H2O2 in chloroplast”. Please provide a reference of this role in chloroplast.

Shigeto, J., Tsutsumi, Y., 2016. Diverse functions and reactions of class III peroxidases. New Phytol. 209, 1395–1402. doi:10.1111/nph.13738

Passardi, F., Cosio, C., Penel, C., Dunand, C., 2005. Peroxidases have more functions than a Swiss army knife. Plant Cell Rep. 24, 255–65. doi:10.1007/s00299-005-0972-6

Almagro, L., Gómez Ros, L. V, Belchi-Navarro, S., Bru, R., Ros Barceló, A., Pedreño, M.A., 2009. Class III peroxidases in plant defence reactions. J. Exp. Bot. 60, 377–90. doi:10.1093/jxb/ern277

GST

In line it is stated that “GST reduces hydrogen peroxide (H2O2) in fatty acids and nucleic acids to prevent ROS accumulation”. In plants, glutathione transferases (GSTs, EC 2.5.1.18) are multifunctional enzymes included in different classes (Phi, Tau, Zeta, and Theta clases, among others). GST have important roles in cellular detoxification of xenobiotics protection against oxidative stress as well as diverse ligand-binding activities (doi: 10.1104/pp.112.205815)

SOD activity. Please, specify the method used for measuring SOD activity [ferricytochrome c and xanthine/xanthine oxidase as the source of O2._ radicals (McCord and Fridovich, 1969)?]. I cannot find the references 30 and 31. Did the author measure the level of proteins? It is important to know to what extent salt and ACC treatments affect the protein levels and to know the specific activity of SOD. In Fig. 5E, Y-axis legend “SOD activity (U/g)” must be “SOD activity (U/g FW)”. The same for Fig. 5G.

  1. Ma, Y.T.; Mi, M.; Bai, M.M.; Ma, H.L.; Liu, X.N.; Yu, J.H.;Shao, H.L. Effects of CPT on seed germination and seedling MDA content of alfalfa. J Agr Sci 2020, 48(8), 1-5. 639
  2. He, B.; Ye, H.B.;Yang, X.E. Effects of Pb on chlorophyll contents and antioxidant enzyme activity in leaf for Pb-accumulating and non-accumulating ecotypes of Sedum. J Environ Sci 2003, 22(3), 274-278.

Y-axis scaling. I think is better to start Y-axis at 0 and to include a break to illustrate the differences.

Fig. 5. It should be nice to see a photo of the quinoa plants. I think that this photo can be included in Fig.5ª.

Fig. 4. Quantitative real time PCR (qRT-PCR) analysis. Why did the authors carry out a qRT-PCR at 0, 3, 6, 9, 12, 24, and 36 h?. Validation of the RNA sequencing analysis data by quantitative real-time PCR (qRT-PCR) analysis are carried out using the same RNA samples

Please, provide the method was used for the determination of gene expression, (Pffafle or Livak etc.) in Mat&Met section.

Wording and grammar. A thorough going-over is required, preferably by a native English speaker, to make this manuscript publication worthy in its language. For example:

Line 73. Eliminate “The”

Line 89: “was reported to play”

Line 91 “were proved to confer”

Line 357 Verification of RNA-seq ”results”

Line 419 The ERFs, ARF, ABP, PYL4 and JIP are in plant hormone signaling, …

Round 2

Reviewer 2 Report

Manuscript ID: plants-1403961

Genome-wide transcriptomic and proteomic ex ploration of molecular regulations in quinoa responses to ethylene and salt stress

I’ve read the revised version of the ms. In my opinion the authors did not address the main points hightlighted in my review.

1.- Please, describe the relevance of this work, the novelty. Authors have a huge amount of data, but the discussion and the relevance of the work are poorly defined.

  1. The difference between the transcriptional reprogramming of genes under high salinity and under high salinity + ACC treatments must be clearly shown. For example, using log2 fold changes (or FPKM values) and carried out a heat map with hierarchical clustering (the same for proteomic analysis). Then, a sound discussion comparing the results of this work with previous studies regarding genes and proteins induced by salt and/or ACC in quinoa or in model plant species both halophytes and glycophytes must be included.

For example, the discussion section can be started with the two components of salinity stress in plants: nonspecific osmotic stress that causes water deficit and, specific ion effects that provoke the accumulation of toxic ions. Moreover, it is also important to talk about the genes activated after sensing the salt signal and, to what extent these responses are similar or different in salt+ACC treatments. 

In this section, in general, authors started providing a general description, many times with no connection with the results obtained, provide a list of genes (a very long list), and finished with the same statement “further study is needed”. For example, osmotic adjustment lines 510-521 “It was reported that exclusion of Na+ from the  cytoplasm is primarily encoded by salt overly sensitive 1 (SOS1) and Na+/H+ exchanger 1 (NHX1). The SOS1 is a Na+/H+ antiporter, which transfers Na+ to the apoplast, while the  NHX1 sequesters Na+ inside the vacuole to decrease the Na+ in the cytoplasm [41-42]. The expression of CqSOS1A, CqSOS1B and CqNHX1 were up-regulated under the salt treatment [12, 34, 43]”, but no SOS or NHX genes are described in their results.

Please, rewrite this section following a logic order and focusing in the differences between salt and salt +acc treatment.

The same for the transcripts found to be involved in plant hormone signal transduction, or genes involved in the synthesis of specific secondary metabolites (betalain and flavonoids). In the literature there are updated reviews of the role of these compounds in plant salt tolerance mechanisms.

Moreover, the correlation between transcriptomic and proteomic results required a throught revision. It is quite difficult to understand.

Data obtanined in Fig 4 and Fig 5 are not discussed. For example, the expression of GLC gene is upregulated in salt+ACC treatment from 6 to 36 h, or the highest content of soluble sugar content found under salt treatment alone.

3. The use of udated literature is recommended. For example, compare your results with this article doi: 10.3390/genes10121042. As far as the role of betalain and salt tolerance, see https://doi.org/10.3389/fpls.2021.683891.

Other comments

Wording and grammar. A thorough going-over is required, preferably by a native English speaker, to make this manuscript publication worthy in its language.

Lines 472-3: What is the meaning of  “The ERFs, ARF, ABP, PYL4 and JIP are in plant hormone responses”?

Line 477: What is the meaning of  “ROS scavenging enzymes function in quinoa responses to ethylene and salt stress”? Do you mean:” upregulated”

Line 479: Provide a reference “Salt stress causes ROS accumulation and oxidative stress aggravation”

Line 480-481. Provide an updated description of the role of antioxidant enzyme

Line 558. Lignins, pectins, celluloses and hemicelluloses are not structural proteins.

Line 560: “Transcriptional changes of the genes in cell wall organization were detected after salt treatment of quinoa seedlings” . Transcriptional changes of the genes involved in

Line 561: The genes in suberin and cutin biosynthesis, photosynthesis and chloroplast were significantly changed. Rewording. The genes involved in

Line 573: “4.5. Secondary metabolism associated proteins s” . Secondary metabolism-associated proteins

Lines 575-577. “Betalain, a tyrosine-derived, red-violet and yellow pigment in quinoa with antioxidant activity, plays important roles in salt response, and CqCYP76AD1-1 was reported in the betalain biosynthesis process in quinoa”. Rewording. Split this long sentence.

Line 604 “4.6. Correlation between the proteomic result and transcriptomic data” 4.6. Correlation between transcriptomic and proteomic data

Line 605: What is the meaning of  “Depending on the different activation methods in the presence of stimuli”?

Round 3

Reviewer 2 Report

I've just read the new version of the manuscript that you sent to me. My decision iS ACCEPTED.

Author Response

We had E-mail our revised manuscript to Editor and Editor gave our comments of Reviewer, which we answered together with Editor's question.